# Determination of Meteor Vector Velocity Using MU Interferometry Measurements of Head Echoes

**Xin Xie [1], Zhangyou Chen [1,*], Li Wang [2] , Heng Zhou [1] and Xiongbin Wu [1]**

[1] Electronic Information School, Wuhan University, Wuhan 430072, China; xinxie@whu.edu.cn (X.X.); hzhou@whu.edu.cn (H.Z.); xbwu@whu.edu.cn (X.W.)

[2] College of Meteorology and Oceanography, National University of Defense Technology, Changsha 410073, China; warren@whu.edu.cn

[*] Correspondence: chzhyou@whu.edu.cn; Tel.: +86-135-4524-3963

**Abstract:** A new method for measuring the vector velocity of meteoroids using meteor head echoes is proposed in this study. The lateral velocity is determined by utilizing the phase interference measurement between channels, while the radial velocity is obtained using a conventional Doppler frequency shift measurement. Compared to previous studies, this method does not require multi-site observations and can calculate the vector velocity of meteors in real-time. This paper provides the complete process for the inversion of the meteor vector velocity, detailing the analyzing process using MU radar head echo data. First, the MUSIC algorithm was used to estimate the DOA of the meteor target, which is a parameter required for lateral velocity measurement. Channel calibration is required before this estimation. Next, delay-Doppler matched filter processing was performed on each receiving channel's data to determine the distance and radial velocity of the meteor target. Subsequently, the lateral velocity component was synthesized using the least squares method from the phase difference rate extracted from the matched filter output results of multiple channel pairs. Then, the vector velocity and trajectory of the meteor could be determined. The method was verified using MU radar head echo data. Different groups of channel pairs were selected for calculating the lateral velocity, and the results were found to be close, demonstrating the self-consistency of the method. Additionally, the calculated vector velocity is consistent with the direction and magnitude of the meteor's motion trajectory, confirming the feasibility of the proposed approach. The method allows for the observation of more prominent characteristics of meteoroid motion, providing a more detailed observation capability of velocity variations in other directions than previous methods.

**Keywords:** vector velocity of meteoroid; MU radar; interference measurement; phase difference rate; matched filter





## 1. Introduction

As meteoroids enter the Earth's atmosphere at a sufficient speed (about 11–72 km/s), they collide with atmospheric molecules, causing rapid heating and ionization of the surrounding neutral air, resulting in the formation of plasma. This process usually occurs at an altitude between 70 and 140 km above the Earth's surface [1,2]. The ionized plasma generated by meteors can be divided into two distinct states: one is the dense and instantaneous plasma region that moves with the ablation of the meteoroids, and the other is the diffusive plasma region that remains in the atmosphere and moves with the neutral winds. Both of these plasmas reflect radio waves when measured with radar, which can create the so-called meteor head and meteor trail echoes [3,4].

The meteor head is a high-density plasma region that surrounds the meteoroid, and the radar cross-section (RCS) of this target is much smaller than that of more common meteor trail echoes [2,5], requiring specific high-power large-aperture (HPLA) radar facilities for observation, e.g., the Middle Atmosphere Alomar Radar System (MAARSY), the Middle and Upper Atmosphere (MU) Radar, the Equatorial Atmosphere Radar (EAR), the Jicamarca

Radio Observatory (JRO), the Poker Flat Incoherent Scatter Radar (PFISR), the European Incoherent Scatter radar system (EISCAT), the Poker Flat Incoherent Scatter Radar (PFISR), and the Advanced Research Projects Agency Long-Range Tracking and Instrumentation Radar (ALTAIR) [2]. HPLA radars can effectively detect and quantify a large number of meteor head echoes. Observing meteor heads is important for several reasons. First, it provides a unique perspective to study the motion and abundance of small bodies in the solar system. The measurements of meteor head echoes offer information about the physical processes that occur during ablation and ionization, providing insight into how meteors evolve over time [6,7]. Additionally, observing head echoes can serve as a probe for studying the upper atmosphere. While atmospheric balloons can measure the high atmosphere, observing atmospheric drag on near-Earth satellites can be studied in the low atmosphere [8]. Importantly, the velocity of meteors is directly related to their equivalent orbits around the sun, providing an astronomical background for measuring atmospheric meteors. Furthermore, measuring the velocities of meteoroids in the Earth's atmosphere is important for understanding the solar system dynamics of small interplanetary bodies. The velocity distribution of meteoroids is a primary constraint for dust generation models in the solar system [9] and is crucial for estimating atmospheric dust input [10,11] and assessing models of meteoroid impact risk to spacecraft [12]. Therefore, accurately measuring meteor velocities is essential to the research above.

Measurements of head echoes provide accurate radial velocities (along the radar line of sight). Currently, most meteor velocity detection is achieved through radial velocity detection, with fewer methods for detecting the vector velocity of meteors. The current research methods are as follows: the first method is multi-station measurements of meteor head echoes. This allows for the measurement of three independent components of the meteoroid's velocity. Common volume observations are used to determine the meteor's position and absolute velocity. An example of this method is the tristatic EISCAT radar, which includes three fully steerable parabolic antennas. A transmitter/receiver is located near Tromsø, Norway (69.59°N, 19.23°E), with two remote receivers located at Kiruna, Sweden (67.86°N, 20.44°E) and Sodankylä, Finland (67.36°N, 26.63°E). Three receivers can be used for tristatic observations together [13,14]. The second method is single-station interferometric measurements of meteor head echoes. This method determines the position of the meteor target based on the radar pulse received by each channel and is used to convert the accurately measured radial velocity component of the meteoroid into the vector velocity. This transformation provides accurate radiation localization and meteoroid velocity determination but depends on an accurate trajectory estimate of the target's lateral motion, which is never as precise as the radial velocity component [15]. Currently, this method is typified by the Jicamarca radar [16,17].

In this paper, we propose a new method for measuring the vector velocity of meteors using a single-station interferometric MU radar. With this approach, a single radar station can successfully obtain the velocity vector without relying on the accurate estimation of the distance to each radar receiving station. Furthermore, this method is cost-effective, computationally simple and easy to implement. Compared to multi-station radars, this method does not require real-time and location coordination, avoiding errors and time delays in the calculation of velocity vectors. Therefore, the method offers high accuracy and real-time measurement capabilities.

The paper is organized as follows. First, we describe the experimental setup. Next, we explain how to use the interferometric method to measure the lateral velocity components and how to synthesize the velocity vector. We then present the process for measuring the radar experimental parameters and display the observation results for a meteor event, including the use of the MUSIC algorithm to determine the instantaneous direction of arrival (DOA) of the target in an inter-pulse period (IPP), the use of a matched filtering technique to obtain the distance and Doppler velocity of the meteor target, the lateral velocity measurement method, and the combination of the estimated distance data and the

radial velocity to determine the meteor trajectory and vector velocity. Finally, we conclude and discuss future potential applications of the proposed methods.

## 2. Experimental Setup

The MU radar was built in 1984 by the Radio Science Center for Space and Atmosphere, RASC (now the Research Institute for Sustainable Humanosphere, RISH) of Kyoto University in Shigaraki, Shiga, Japan (34.85°N, 136.11°E). The radar is primarily used to study the atmospheric and plasma dynamics over a wide range of altitudes from the troposphere to the ionosphere. The MU radar is a powerful single-station pulse Doppler radar operating at a frequency of 46.5 MHz, including a 25-channel digital receiver system where each digital channel outputs the sum of the radio signals received from 19 sub-groups of Yagi antennas. The entire array consists of 475 antennas uniformly distributed within a circular aperture of 103 m. The peak and average output powers are 1 MW and 50 kW, respectively, and the antenna beam is conical with a round-trip (two-way) half-power beam width of 2.6 degrees [18]. Figure 1 illustrates the array and sub-group configurations and its schematic diagram of operation.

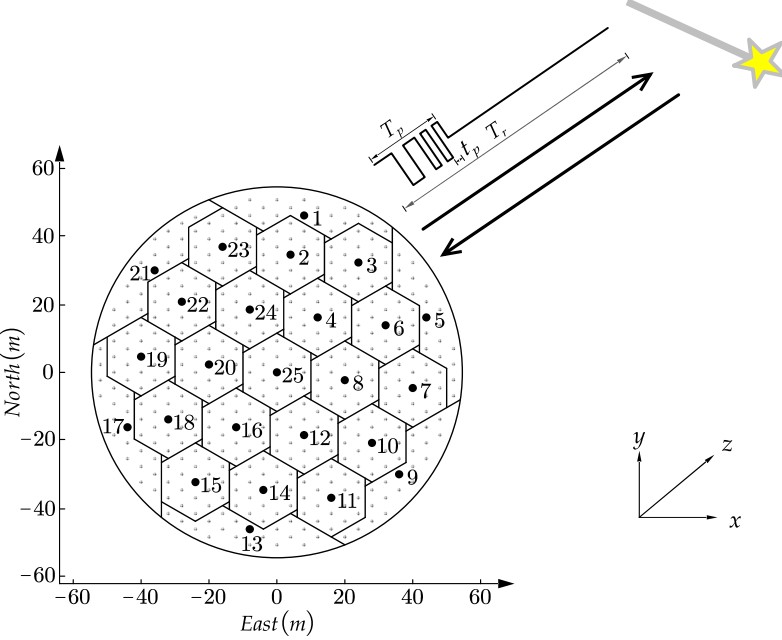

**Figure 1.** Schematic diagram of MU radar antenna array and transmitting and receiving echo. The MU radar consists of 475 antennas arranged in an equilateral triangular grid with a unit spacing of 0.7 [18]. The array is divided into 25 sub-groups, each containing 19 antennas connected to their respective transceiver module, forming the main beam that is transmitted and received by different channels. The origin of the coordinate system is at the center of the MU radar (also the location of channel 25), the *x*-axis points east, the *y*-axis points north, and the *z*-direction is towards the zenith angle.

The MU radar underwent an upgrade in March 2004, which included 25 digital receivers and has allowed for more extensive observation of meteors and other solid targets. Since 2009, regular meteor observations have been conducted using the upgraded 25-channel digital receiver system and the new head echo mode [15,19]. Table 1 provides the main radar parameters of the MU radar in head echo mode for meteor observations. In head echo mode, a phase-coded pulse waveform is transmitted with a 13-bit Barker code (12 μs baud length) used for phase coding, and the pulse length is 156 μs, available at a full duty cycle of 5%. In reception, a sampling period of 6 μs is used, and all 25 channels of the MU radar are oversampled by a factor of two. The main pulse sequence echoes received

are from the height range of 70–130 km, which is the most important part of the meteor region and where the majority of head echoes occur [19].

**Table 1.** MU radar's parameters for head echo mode.

| Parameters | Values |
| --- | --- |
| Latitude | 34.85°N |
| Longitude | 136.10°E |
| Frequency | 46.5 MHz |
| Beam full width at half-maximum (FWHM) | 3.6° |
| Pulse code | 13-bit Barker Code |
| Inter-pulse period (IPP) | 3.12 ms |
| Pulse width | 156 μs |
| Baud length | 12 μs |
| Sample rate | 2 |
| Range gate | 900 m |
| Sample range | ≈73–127 km |

The radar data used in this article was collected in June 2018 from the MU radar system released by the RISH. The time period for the data is from 08:00 on 27 June 2018 to 08:00 on 29 June 2018 (UTC + 8/JST).

## 3. Method of Measuring Vector Velocity

In a conventional interferometric HPLA radar system, head echo observations cannot directly measure the vector velocity of meteors. Typically, only radial velocities, which refer to the motion velocities of targets toward or away from the radar, can be estimated. Radial velocities cause a frequency shift of the received signals relative to the carrier frequency, known as the Doppler shift ($f_d$). This shift serves as the basis for most radar velocity measurements and can be written as follows:

$$f_d = \frac{2f_0 v_r}{c} = \frac{2v_r}{\lambda},\tag{1}$$

where $f_0$ is the carrier frequency, $c$ is the speed of light, and $\lambda$ represents the wavelength. Therefore, the estimated value of the radial velocity ($v_r$) of a meteor can, in principle, be derived from the $f_d$ of a single received radar pulse.

This article focuses on the measurement of the vector velocity, which is split into the radial velocity and lateral velocity. The radial velocity represents the component of the velocity along the line of sight, denoted as $\vec{v}_r$ and its magnitude is determined by Equation (1). The lateral velocity, represented by $\vec{v}_\eta$, represents the component of the velocity perpendicular to the line of sight. Therefore, the vector velocity can be expressed as the sum of the lateral velocity and radial velocity, as shown below:

$$\vec{v} = \vec{v}_\eta + \vec{v}_r\tag{2}$$

Next, how to solve for $\vec{v}_\eta$ will be discussed. Starting from the motion of a scattering point, we first introduce the method of measuring two-dimensional (2D) lateral velocity. Then, we derive the method of measuring three-dimensional (3D) lateral velocity in space.

### 3.1. The 2D Velocities Measurement

Two channels are needed to form a basic interferometer to measure the lateral velocity of an object. Figure 2 illustrates the geometric structure of this measurement system in two dimensions. Two receiving antennas with a distance of $D$ between them are used to observe a target scattering in the far field [20].

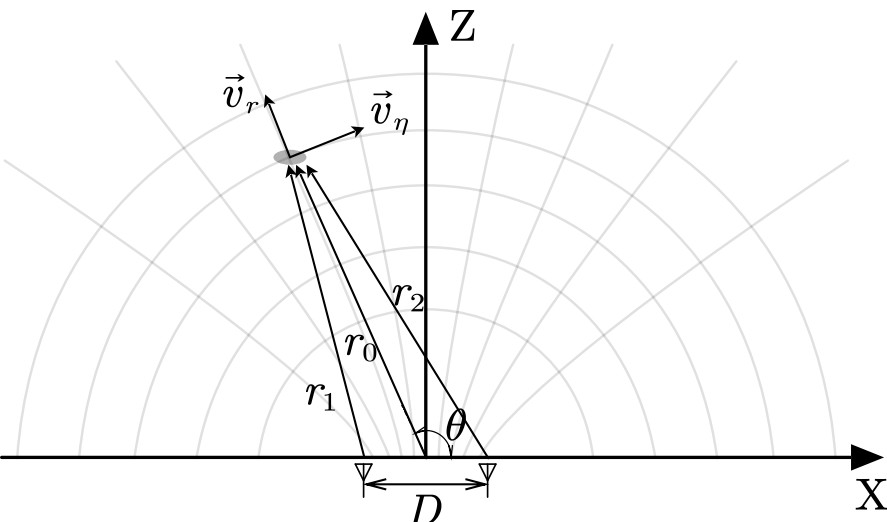

**Figure 2.** Array diagram for measuring two-dimensional lateral velocity. The vector velocity is decomposed into two components.

Assume that the radio waves with angular frequency $\omega$ and wavelength $\lambda$ are scattered by a moving object, neglecting the initial phase factor. The expressions for the signals received by antennas 1 and 2 are given by the following:

$$s_1 = \exp\left[j\left(\omega t - \frac{2\pi}{\lambda}r_1\right)\right] \tag{3}$$

$$s_2 = \exp\left[j\left(\omega t - \frac{2\pi}{\lambda}r_2\right)\right] \tag{4}$$

Moreover, the corresponding differential signal can be expressed as follows:

$$s_\Delta = s_1^* \times s_2 = \exp\left[\frac{2\pi}{\lambda}(r_2 - r_1)\right] \tag{5}$$

In addition, the phase component of the differential signal is defined as follows:

$$\varphi_\Delta = \frac{2\pi}{\lambda}(r_2 - r_1) \tag{6}$$

Figure 2 depicts an orthogonal locus curve. It is evident that when the moving object travels along the hyperbolic envelope, with its two foci located at $(-D/2, 0)$ and $(D/2, 0)$, the difference between $r_2$ and $r_1$ remains constant, resulting in the phase component remaining unchanged. This motion does not cause any change to the envelope. If the moving object is travelling together with the ellipse, then the differential signal envelope will have the maximum change, since the ellipse and hyperbola are orthogonal, as long as they share the same foci. When $r_0 >> D/2$, the motion along the ellipse envelope and the hyperbola envelope corresponds to the lateral velocity and radial velocity, respectively; under this condition, the following can be proven:

$$k_p = \frac{2\pi D}{\lambda r_0} v_\eta \sin\theta_0 \tag{7}$$

Here, $k_p$ represents the rate of phase change of the envelope signal caused by lateral motion, as shown in Figure 2. This parameter can enable the measurement of the lateral velocity $v_\eta$ of a stationary target located at distance $r_0$ and zenith angle $\theta_0$. Li et al. [21] and Wang et al. [22] validated this method through both theoretical simulation and practical experiments.

### 3.2. The 3D Velocities Measurement

To extend the measurement of 2D lateral velocity in 3D scenarios, we considered a target in the 3D space. To achieve this, multiple pairs of interferometric antennas are required to measure different lateral velocity components, which are subsequently synthesized into a vector velocity. Figure 3 illustrates how to measure this 3D lateral velocity of a meteor using two pairs of antennas to obtain two lateral velocities.

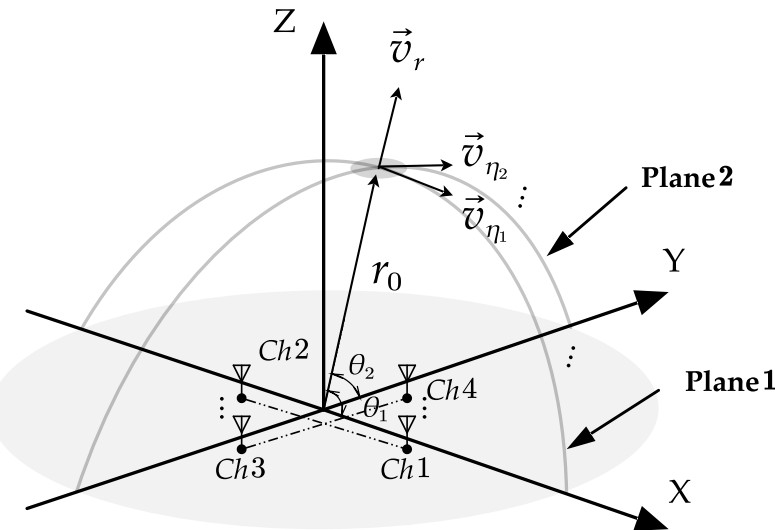

**Figure 3.** The geometry of the receiving antennas for 3D lateral velocity measurement (only two pairs of antennas are shown for clarity). Plane 1 and Plane 2 refer to the planes containing the target, the center of the antenna, and the corresponding lateral velocity components. Multiple lateral velocity components can be included.

Based on the analysis of the 2D case, lateral velocity $v_{\eta 1}$ can be obtained through the signal from the antenna pair on the *x*-axis, while lateral velocity $v_{\eta 2}$ can be obtained through the antenna pair on the *y*-axis. Following the approach presented in this paper, the vector velocity is decomposed into two mutually orthogonal components: radial velocities $\vec{v}_r$ and lateral velocities $\vec{v}_\eta$. It should be noted that the lateral velocity is the total lateral velocity and lies in the same plane as the multiple lateral velocity components $\vec{v}_{\eta_i}(i = 1, 2, \ldots, n)$ shown in Figure 3. The two lateral velocity components (for simplicity, only two are shown) can be understood as the projection components of the total lateral velocity onto Planes 1 and 2. Based on the dot product operation and the vector projection relationship, the lateral velocity component and the total lateral velocity satisfy the following relation:

$$\vec{v}_\eta \cdot \vec{v}_{\eta_i} = \left|\vec{v}_{\eta_i}\right|^2 \cdots i = 1, 2, \ldots, n \tag{8}$$

The vector coordinate representation of lateral velocity is given by the following:

$$\vec{v}_\eta = \left(v_{\eta_x}, v_{\eta_y}, v_{\eta_z}\right), \tag{9}$$

$$\vec{v}_{\eta_i} = \left(v_{\eta_{ix}}, v_{\eta_{iy}}, v_{\eta_{iz}}\right) \, i = 1, 2, \ldots, n, \tag{10}$$

where $\vec{v}_{\eta_{ix}}$, $\vec{v}_{\eta_{iy}}$, and $\vec{v}_{\eta_{iz}}$ represent the three components of $\vec{v}_{\eta_i}$ in the Cartesian coordinate system.

By substituting Equation (9) into Equation (8), we obtain

$$\begin{bmatrix} v_{\eta_1 x} & v_{\eta_1 y} & v_{\eta_1 z} \\ v_{\eta_2 x} & v_{\eta_2 y} & v_{\eta_2 z} \\ v_{\eta_n x} & v_{\eta_n y} & v_{\eta_n z} \end{bmatrix} \begin{bmatrix} v_{\eta x} \\ v_{\eta y} \\ v_{\eta z} \end{bmatrix} = \begin{bmatrix} \left| \vec{v}_{\eta_1} \right|^2 \\ \left| \vec{v}_{\eta_2} \right|^2 \\ \left| \vec{v}_{\eta_n} \right|^2 \end{bmatrix}. \tag{11}$$

Equation (10) can be expressed as **Ax = b**, where **b** is the squared magnitude of the vector, **A** is the matrix represented by the rectangular coordinates of the lateral velocity components, and **x** is the synthesized lateral velocity vector. The least-squares method can be used to solve for the lateral velocities from multiple pairs of antennas. The least-squares solution of Equation (11) is

$$\mathbf{x} = \begin{bmatrix} v_{\eta x} \\ v_{\eta y} \\ v_{\eta z} \end{bmatrix} = \left( \mathbf{A}^T \mathbf{A} \right)^{-1} \mathbf{A}^T \mathbf{b}. \tag{12}$$

Then, the velocity can be calculated using Equation (2).

Due to the multiple symmetric antenna pairs of the MU radar, not only the radial velocities, but also the lateral velocities can be measured. The measurement of the lateral velocities is accomplished via the interferometry technique, which has the main advantage of measuring motions that cannot be detected by Doppler sensors. When the radial motion is minimized or becomes zero, Doppler sensors are unable to identify the motion of objects. However, when the motion is strictly lateral (zero radial velocity), the interferometric antenna pairs can measure the highest frequency shift [23]. Therefore, the proposed approach in this paper can enable a single radar station to successfully obtain the velocity vector.

## 4. Signal Process and Results

Accurate vector velocity measurement requires the joint efforts of various signal processing modules, as illustrated in Figure 4. After the MU radar data is received by 25 channels, the calibration of each channel is necessary to offset any possible phase and amplitude errors. The MUSIC algorithm is employed to estimate the DOA for angle estimation, fully utilizing the advantage of an array's large aperture to achieve higher angular resolution, which is crucial for determining the direction of the final vector velocity. In addition, considering the high Doppler shift characteristics of meteor targets, a delay-Doppler matched filter is constructed to obtain the radial velocity and line-of-sight distance of the meteor. The phase of the matched filter output is then extracted, and multiple pairs of channels are interfered to obtain the phase difference change curve. Due to the presence of phase wrapping, phase unwrapping is required prior to obtaining accurate phase differences. Afterwards, the lateral velocity was calculated by using the proposed lateral velocity calculation method in this study. Finally, the results obtained from each signal processing module are jointly used to reconstruct the meteor's trajectory and infer its vector velocity.

### 4.1. DOA Determination and Channel Calibration

Meteoroids and hard targets produce coherent radar echoes, which can be used to determine the location of the target by finding the DOA of the incident radar echoes. In this study, the multiple signal classification (MUSIC) algorithm developed by Schmidt [24] was used to estimate the DOA from the head echo of the MU radar. Considering the MU radar planar array shown in Figure 1, the received signal model is represented as follows:

$$\mathbf{X}(t) = \mathbf{A}(\theta, \varphi)\mathbf{S}(t) + \boldsymbol{\xi} \tag{13}$$

where $\mathbf{X}(t)$ is the $M \times 1$ snapshot data vector of the array, $\boldsymbol{\xi}$ is a complex Gaussian white additive noise data vector of size $N \times 1$, and S(t) is the complex envelope $N \times 1$ data

vector of the signal. $\mathbf{A}(\theta, \varphi) = [a(\theta_1, \varphi_1), a(\theta_2, \varphi_2), \ldots, a(\theta_N, \varphi_N)]$ is an $M \times N$ steering vector matrix, where $(\theta, \varphi)$ is the DOA of the ith source, corresponding to its azimuth and elevation angles. $a(\theta, \varphi)$ is the standard steering vector, which is given by the following equation:

$$a(\theta, \varphi) = g(\theta, \varphi) \odot \exp\left(-jr^T k\right). \tag{14}$$

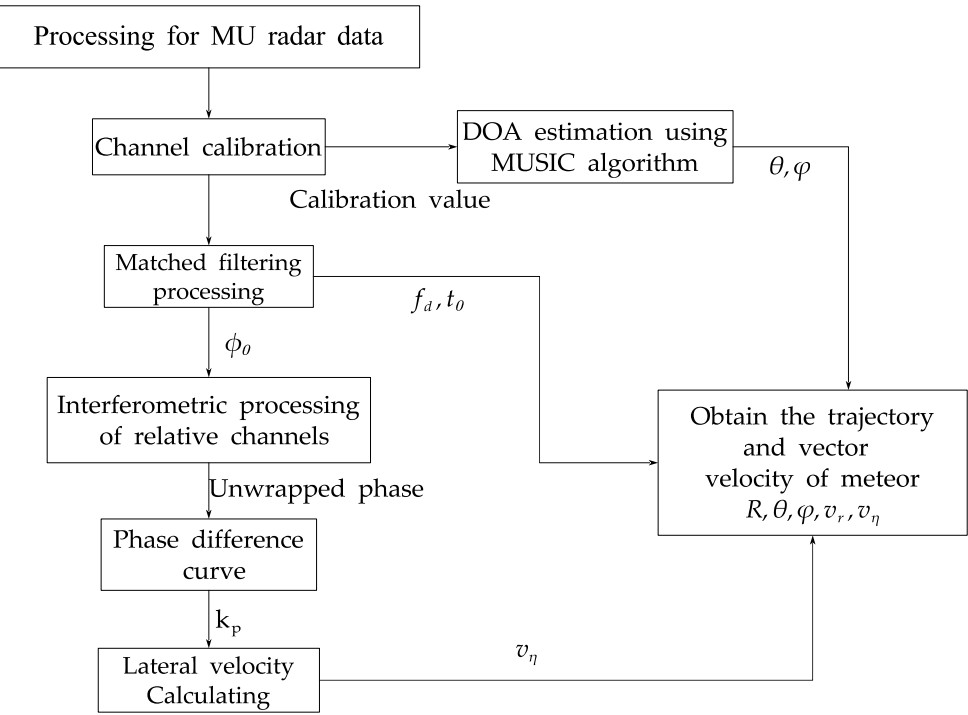

**Figure 4.** Diagram of the signal process: $\phi_0$ is the phase of channel matched filter output, $R$ is the line-of-sight distance, and $\theta$ and $\varphi$ are the azimuth and elevation angle, respectively.

Here, $r = [r_x, r_y, r_z]^T$ (a $3 \times M$ matrix) represents the center positions of the antenna subarray with respect to the geometric center of the entire array, given in terms of the radar wavelength. $k = 2\pi[\cos\varphi\sin\theta, \cos\varphi\cos\theta, \sin\varphi]$ is the wavenumber vector (a $K \times 1$ complex vector), and $\odot$ denotes the Hadamard product; $g$ is the gain patterns for the transmitting and receiving antennas [19]. In the MU radar, the entire system uses the same type of Yagi antenna, which allows for a common function to be used for all the antennas [18]. This means that the dependency of the subarray gain on the input DOA is the same, and thus can be omitted (normalized to 1). In addition, for the horizontally oriented MU radar antenna, the value of $r_z$ is zero, which allows Equation (14) to be simplified as follows:

$$a(\theta, \varphi) \cong \exp\left(-2\pi j\left(r_x \cos\varphi \sin\theta + r_y \cos\varphi \cos\theta\right)\right). \tag{15}$$

The spectral estimation formula for the MUSIC algorithm is given by

$$P(\theta, \varphi) = \frac{1}{a^H(\theta, \varphi) U_N U_N^H a(\theta, \varphi)}, \tag{16}$$

where $H$ is the Hermitian transpose and $U_N$ is the noise subspace obtained by performing eigendecomposition $[U, \Sigma] = \text{eig}(R)$ on the covariance matrix of the received signals, and then selecting the eigenvectors corresponding to the smaller eigenvalues. The number of larger eigenvalues can be obtained through methods for estimating the number of signal sources. Therefore, the remaining eigenvalues correspond to the smaller eigenvalues. The received data covariance matrix can be expressed as $R = \frac{1}{M} XX^H$. When evaluating Equation (16) for different DOAs, the denominator approaches zero near the DOA of the

signal, and this results in a narrow peak in the spectrum $P(\theta, \varphi)$. The DOA estimation can then be achieved through angle searching.

In practice, the parameters of the electronic components in antenna systems may vary due to factors such as temperature, time drift, or aging effects, resulting in errors in the phase and amplitude of the array elements. Therefore, it is necessary to perform regular array calibration during the operation of these antennas to maintain their original performance. In the presence of amplitude and phase errors, the array steering vector requires correction as follows:

$$\widetilde{\mathbf{A}}(\theta, \varphi) = \mathbf{\Gamma}\mathbf{A}(\theta, \varphi), \tag{17}$$

where $\mathbf{\Gamma}$ is the phase error matrix as

$$\mathbf{\Gamma} = \mathrm{diag}[\Gamma_1, \Gamma_2, \cdots, \Gamma_M], \cdot \Gamma_m = \rho_m \exp(j\,\phi_m), m = 1, 2, \ldots, M, \tag{18}$$

with $\rho_m$ and $\phi_m$ as the gain and phase errors of the $m$-th antenna channel, respectively, $\mathrm{diag}(\cdot)$ returns a diagonal matrix whose diagonal equals the input vector. As the first channel serves as the reference channel, we have $\rho_m = 1$ and $\phi_m = 0$.

Using an inaccurate steering vector $\widetilde{\mathbf{A}}(\theta, \varphi)$ for spectrum peak searching may result in peak shifting or indistinguishable peaks. To compensate for array phase errors, the calibration of each receiving channel of the MU radar is required. We should estimate the amplitude and phase errors $\mathbf{\Gamma}$ by a certain method, and then substitute $\mathbf{\Gamma}\mathbf{A}(\theta, \varphi)$ for $\mathbf{A}(\theta, \varphi)$ in Equation (16). Amplitude error estimation can be achieved by computing the signal power of each channel and then normalizing it using a reference channel. The key lies in the estimation of the phase errors. In this study, a statistical method utilizing a large number of strong meteor head echoes was chosen for phase calibration. This method selects multiple strong and well-defined meteor echoes, uses the DOA estimated for each event to generate the best phase estimate through one iteration, and then takes the statistical average of the estimation results for all the strong meteor head echoes [19,25,26]. The 86 strongest meteor head echoes detected by the MU radar from June 28, 2018, 8:00 to June 29, 2018, 8:00 were selected for calibration using this method. The estimation results of the phase errors are shown in Figure 5, revealing a significant error in channel 7.

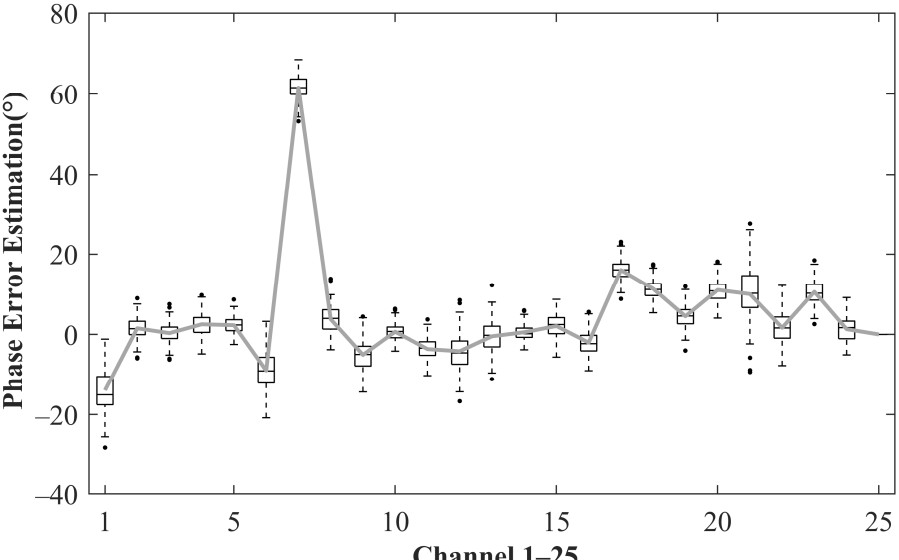

**Figure 5.** Result of phase error estimation. The boxplot was drawn based on the calibration values of all meteor echoes for each channel, with a total of 86 samples. The points on the line are centered on the mean phase error value.

The effectiveness of the method is demonstrated through the DOA estimation results for each head echo event using the calibration values estimated in Figure 5 and substituted

into Equations (17) and (18). An example is shown in Figure 6, where Figure 6a is a range-time-intensity plot of the raw data with 512 IPPs and 85 sampling points. Due to truncation, the distance at the distance element of 1 is about 73 km and should be added in the final distance calculation. The meteor echo event occurred at 08:19:00 on 26 June 2018 (UT + 8/JST), and strong meteor head echoes spanning multiple IPPs were observed during the meteor event, along with multiple distance element migrations, equivalent to a distance of approximately 15 km being crossed in about 0.2 to 0.3 s. Figure 6b is the MUSIC spectrum of the DOA estimate after calibration, viewed from the top angle, showing that the meteor is roughly located directly above, with distinct peaks in the spectrum. To clearly demonstrate the effect of calibration, Figure 7 shows the DOA estimate results before and after calibration for two angular dimensions, indicating a significant improvement in the DOA estimation performance after calibration, as evidenced by the sharper and narrower MUSIC spectral peaks. Remarkably, the presence of five asymmetric array channels in the outermost layer of the MU array results in the absence of a grid lobe near the peak of the MUSIC spectrum. Instead, it appears in the form of sidelobe, as shown in Figure 6b, which can be verified by theoretical calculations.

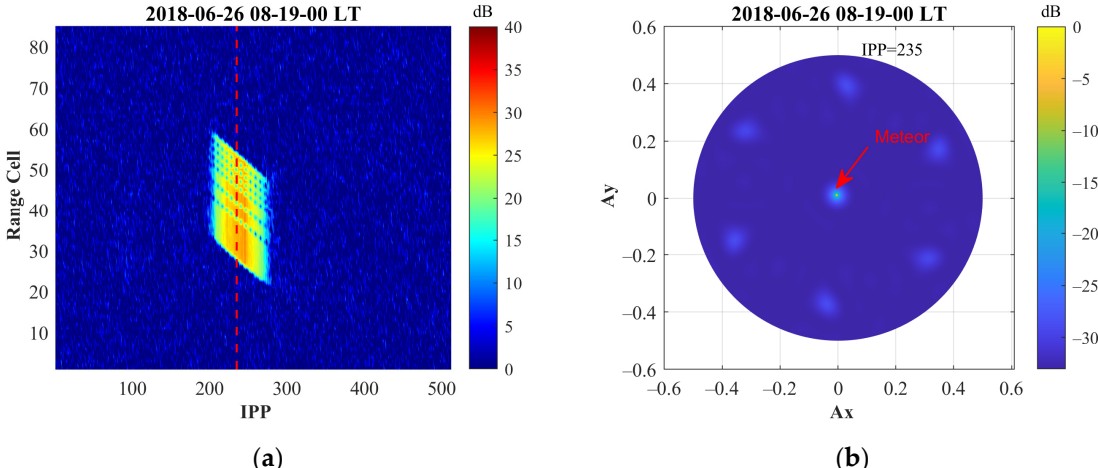

**Figure 6.** (**a**) A range-time-intensity plot of MU radar raw data detected on 19 August 2018 at 19:00 JST. The echo appears as a block of 26 samples, as this is the sample length of the transformed pulse. (**b**) A top view of the 2D MUSIC spatial spectrum of the estimated instantaneous DOA of the meteor after calibration; the meteor appears in an orientation that forms obvious and sharp spectral peaks.

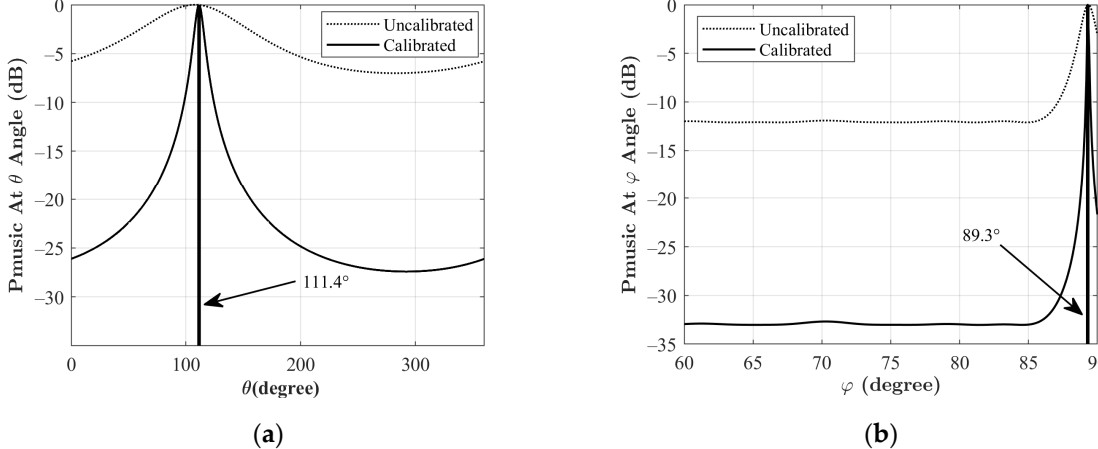

**Figure 7.** Comparison of 2D MUSIC estimation results under two angle dimensions. (**a**) $\theta$ dimension. (**b**) $\varphi$ calibration.

### 4.2. Delay-Doppler Matched Filter

Matched filtering is widely used in various fields of radar signal processing and can be summarized as a technique for filtering signals that match an implemented model. The technique can maximize the signal-to-noise ratio (SNR) of the output signal by computing the cross-correlation function between the received and transmitted signals of the radar.

The meteor head echo comes from the dense region of the plasma that is close to and propagates together with the meteor, resulting in highly transient and Doppler-shifted signals. Because of the large Doppler shift, simple correlation with the transmitted pulse waveform is not sufficient, and correct matching requires Doppler compensation by the radial velocity of the meteor before cross-correlating with the phase-coded pulse transmitted. Next, the matched filtering process of the MU radar received signal is described.

In meteor head mode, the MU radar transmits a pulse waveform encoded by a 13-bit Barker code, which is oversampled by a factor of two. The Barker code with a Baud length of 12 μs is transmitted using a sampling period of $T_s = 6$ μs. Therefore, the Barker code modulation part can be expressed in the form of

$$c_n(n = 0, \ldots, 25) = [1, 1, 1, 1, 1, 1, 1, 1, 1, 1, -1, -1,$$
$$-1, -1, -1, -1, +1, +1, -1, -1, +1, +1, -1, -1, +1, +1], \tag{19}$$

where $n$ represents the nth code element used. Therefore, the transmitted signal can be expressed as

$$\bar{s}(t) = s(t) \exp(j2\pi f_0 t) \tag{20}$$

$$s(t) = \sum_{n=0}^{N-1} \text{rect}\left(\frac{t - nT_s}{T_s}\right) c_n \tag{21}$$

where $f_0$ is the carrier frequency of the transmit signal, $N$ is the number of sampling points calculated under the pulse length, and $s(t)$ represents the phase modulation part of the transmit signal encoded by the Barker code. The horizontal line in $\bar{s}(t)$ denotes that the signal contains a carrier and has not yet been demodulated.

Considering a target moving towards the radar with a radial velocity of $v_r$, the received waveform can be expressed as follows:

$$\bar{s}_r(t) = \sum_{n=0}^{N-1} \text{rect}\left(\frac{t - nT_s - T_r}{T_s}\right) c_n \exp[j2\pi f_0(t - t_r)] \tag{22}$$

$$T_r = \frac{2(R_0 - v_r(t - t_0))}{c} \tag{23}$$

where $R_0$ denotes the initial distance between the target and the radar, and $T_r$ represents the time between the transmission of the radar pulse and its collision with the target. By removing the carrier frequency from the received signal and normalizing it, we can represent it in the following form:

$$s_r(t) \cong \sum_{n=0}^{N-1} \text{rect}\left(\frac{t - nT_s - t_0}{T_s}\right) c_n \exp(j2\pi f_d t) \tag{24}$$

Equation (24) indicates that the received signal incurs an additional Doppler frequency shift compared to the transmitted signal, in addition to introducing a time delay. The Doppler frequency shift, $f_d$, is determined by Equation (1). Considering that meteoroids themselves have high velocities, traditional matched filtering can result in significant mismatch in the output. Therefore, compensating for the Doppler shift during matched

filtering is necessary. Introducing the compensating frequency term, $\exp(-j2\pi f_n t)$, the output of delay-Doppler matched filtering (cross-correlation) can be expressed as follows:

$$\chi(t, f_n) = s_r(t) \exp(-j2\pi f_n t) \otimes s^*(-t), \tag{25}$$

where the symbol * denotes the complex conjugate operation, and the following equation is satisfied:

$$|\chi(t, f_n)| \leq |\chi(T_r, f_d)| = \chi_{\max} \tag{26}$$

Equation (25) represents the output of the matched filtering, which is a bivariate function that peaks at $(T_r, f_d)$, corresponding to the actual target delay relative to the radar and the Doppler frequency shift of the received echo. However, the actual radial velocity of the meteoroid is unknown and needs to be determined. The final distance and radial velocity can be obtained by searching the frequency and selecting the delay and Doppler frequency corresponding to the highest peak of the two-dimensional function. Ideally, after matching, the energy dispersed over the Barker code symbols becomes highly concentrated at the first symbol of the Barker code waveform, thus improving the signal-to-noise ratio of the received signal.

Figure 8 presents an example of the matched filtering results obtained from the head echo data using the selected meteor event from Figure 6a. The matched filtering can provide direct distance and velocity information of the target from each IPP, corresponding to the peak coordinates in the spectrum. As shown in Figure 8a, the maximum output of the matched filter for the 235th IPP is located at (47.95, 28). By performing matched filtering on all the IPPs, the matched filtering RTI plot is obtained, as illustrated in Figure 8b. Next, the variation of distance and velocity across consecutive IPPs is obtained, as illustrated in Figure 9. As shown in the figure, the meteor approaches the Earth at a very high speed, and its radial velocity gradually reduces with time, decreasing from 50 km/s initially to 42 km/s. Additionally, employing the pulse-to-pulse phase matching technique can effectively improve the velocity accuracy after obtaining the Doppler shift of a single IPP echo [25,27]. The technique measures the Doppler shift by utilizing the phase difference between the current IPP and the previous IPP, and the phase unwrapping process for adjacent IPPs relies on the velocity results obtained from the matched filter. The additional lines in Figure 9b show the results obtained through this technique, which are more continuous and stable compared to the results obtained directly.

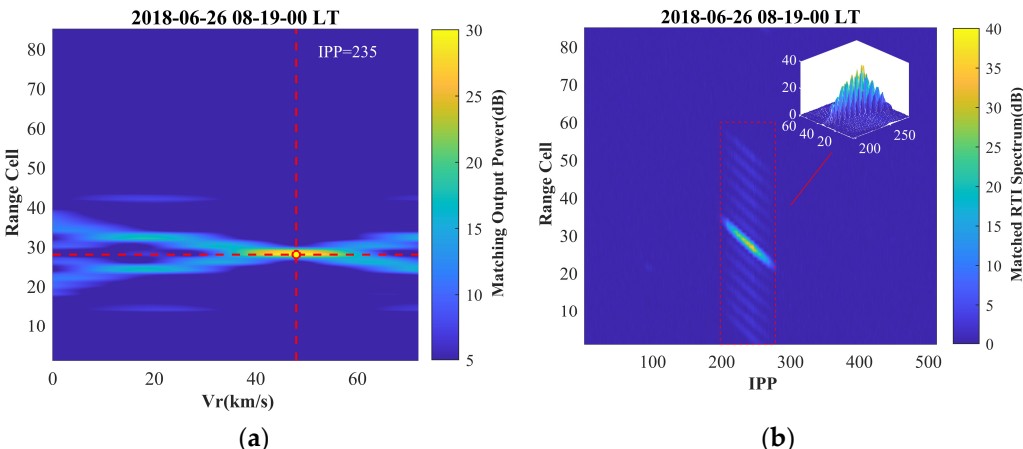

**Figure 8.** The result of the received echo data after being processed with matched filtering. (**a**) The matched filtering results obtained by selecting a specific IPP, with the red line intersection corresponding to the output distance and velocity. (**b**) Range-time-intensity (RTI) plot obtained by matched filtering for all IPPs. The sidelobes resulting from the use of the Barker code still exist in the RTI plot; however, the main lobe is still distinguishable, despite the presence of the sidelobes.

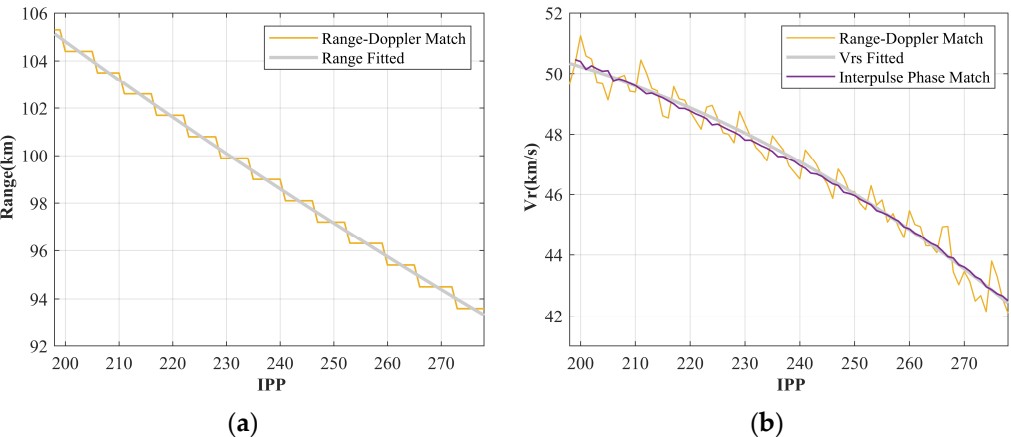

**Figure 9.** The estimated results of distance and velocity are calculated by dashed lines and fitted by solid lines. (**a**) Results for distance. (**b**) Results for radial velocity.

### 4.3. Determination of Lateral Velocity

The determination of the vector velocity is achieved by computing the radial and lateral velocities separately. The radial velocity is calculated using Equation (1) based on the Doppler shift, while the lateral velocity is estimated by utilizing the phase difference of the channel, as shown in Equation (7). The MU radar detects the 3D velocity of the meteor head; therefore, multiple channel pairs need to be selected when calculating the lateral velocity, and subsequently synthesized using Equations (8)–(12).

The selected channel pairs follow the principle of symmetry and minimal distance between each other in order to reduce errors and uncertainty. Based on this principle, we selected two groups of channel pairs, which form the basis of comparison for the final calculation results. The first group comprises channels 3, 7, and 11, and channels 15, 19, and 23, wherein each pair consists of a front and a rear channel; and there is a total of 3 pairs, denoted as Ch3–15, Ch7–19, and Ch11–23. The second group includes channels 2, 6, and 10, and channels 14, 18, and 22, denoted as Ch2–14, Ch6–18, and Ch10–22. Figure 10 shows the selection of these two groups of channel pairs, with each pair connected by a line. Regardless of the choice made, each channel pair can be used to compute a lateral velocity component. The total lateral velocity can be determined by utilizing the lateral velocity components obtained from the remaining channel pairs in the same group.

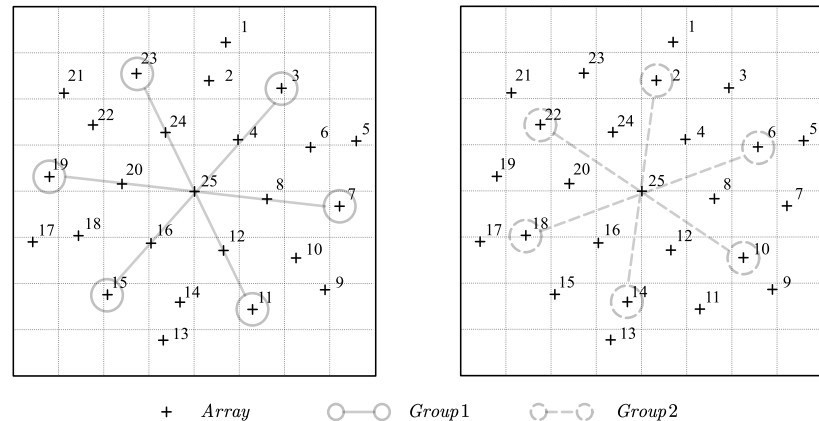

**Figure 10.** Schematic diagram illustrating the selection of channel pairs. Each circular node represents a channel and is connected by a solid or dashed line, indicating a channel pair. Three channel pairs form a group with solid circles representing the front group and dashed circles representing the rear group.

Next, the relative phase difference between the antennas needs to be calculated. The matching filter output is directly used for this purpose, as it provides a certain signal gain. Using Equation (5), the phase difference can be calculated from the differential signal. Ignoring the change of speed in a single IPP time, the relationship between the phase difference and lateral velocity is given by

$$\Delta\phi = k_p T_s = \frac{2\pi D}{\lambda r_0} v_\eta \sin\theta_0 T_s,$$  (27)

where $\Delta\phi$ is the phase difference between adjacent IPPs and Ts is the IPP interval. However, the phase difference needs to be unwrapped to obtain clear results. By substituting the maximum possible lateral velocity into Equation (27), we obtain a $\Delta\phi$ value much smaller than $2\pi$, indicating that the phase difference between adjacent PPIs will be less than $2\pi$. Utilizing this constraint, we can successfully unwrap the phase difference. Figures 11 and 12 show the changes in the continuous PPI phase differences, with the solid lines representing the uncalibrated phase differences and the dashed lines displaying the unwrapped phase difference version. The phase unwrapping is performed by using the sequential point scanning method (SPSM), which sequentially scans the signal from high SNR sampling points toward each end of the signal, and then segments each section and conducts linear fitting within the segment. The resulting phase curve is then adjusted by adding or subtracting a multiple of $2\pi$ from each $\Delta\phi$ to obtain a smooth phase curve. This is accurate for almost all head echoes from MU due to the slow variation of $\Delta\phi$.

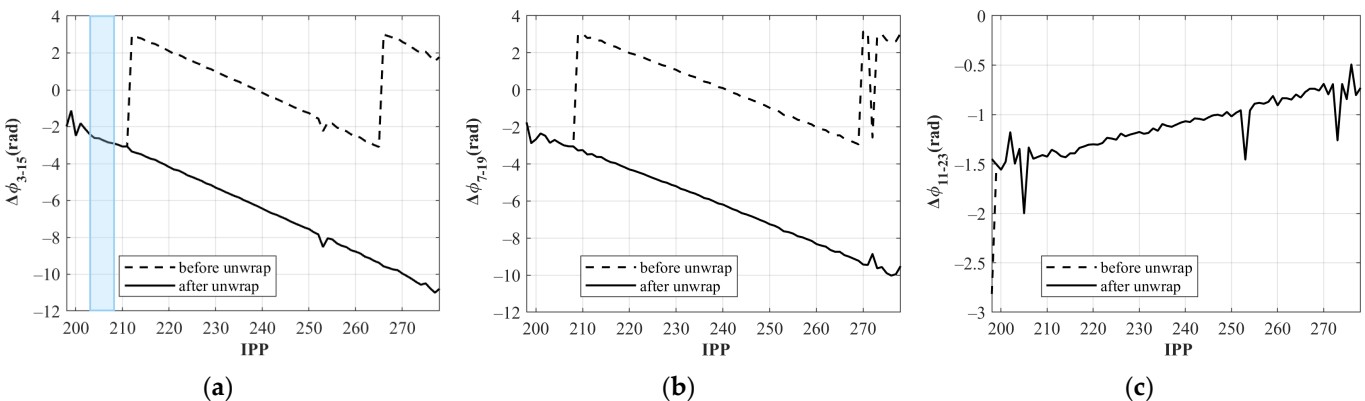

**Figure 11.** Phase difference change of three channel pairs in the first group. (**a**) Results for Ch3–15. (**b**) Results for Ch7–19. (**c**) Results for Ch11–23. The solid and dashed lines represent the unwrapped and uncalibrated phase differences, respectively.

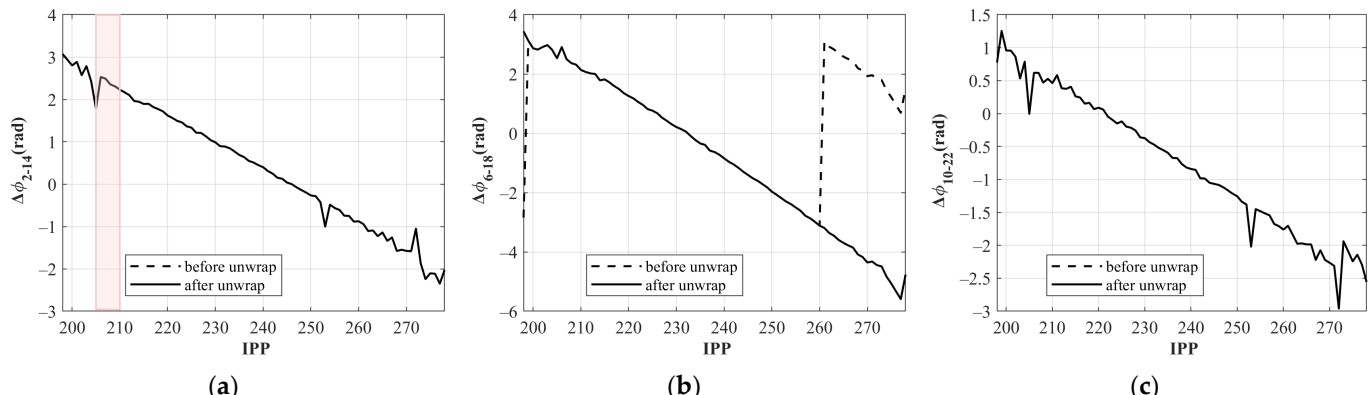

**Figure 12.** Phase difference change of three channel pairs in the second group. (**a**) Results for Ch2–14. (**b**) Results for Ch6–18. (**c**) Results for Ch10–22. The solid and dashed lines represent the unwrapped and uncalibrated phase differences, respectively.

The lateral velocity is determined by calculating the phase difference rate $k_p$ using a simple window fitting method to extract the slope, as described in Equation (7). By scanning the phase difference with a sliding time window, as shown in Figures 11 and 12, the lateral velocities for each channel pair are then calculated using Equation (7) to obtain the values of $\vec{v}_{\eta_{i-j}}$ ($i = 3, 7, 11; j = 15, 19, 23$). Taking the first group as an example, $\vec{v}_{\eta_{3-15}}$ is calculated using the phase difference rate of channels 3 and 15, while $\vec{v}_{\eta_{7-19}}$ and $\vec{v}_{\eta_{11-23}}$ are calculated similarly. Finally, the lateral velocity components are combined and substituted into Equations (8)–(12) to calculate the total lateral velocity. The same approach is applied to the second group.

Figure 13 shows the computed lateral velocity results from the two groups of channel pairs. Figure 13a shows the total lateral velocity magnitude, while Figure 13b–d display the magnitude of the individual lateral velocity components in a Cartesian coordinate system to indicate the direction of the lateral velocity. Choosing multiple groups of comparison results is crucial because it allows us to verify whether this method is self-consistent. The lateral velocity values obtained from the two groups of channel pairs are very close to each other, especially in the middle section where the SNR is the highest, and a consistent trend in the lateral velocity variation can be seen. Combining the signs of $v_{\eta_x}$ and $v_{\eta_y}$, and considering the sign reversal of $v_{\eta_z}$ halfway through, we can infer that the meteoroid passed through the zenith direction and that its trajectory was oriented towards the southwest. Table 2 presents a comparison between Group 1 and Group 2 regarding the total lateral velocity and its three individual components along the x, y, and z axes. It can be observed that the root mean square error (RMSE) of the velocity generally does not exceed 3 km/s, while the mean absolute error (MAE) is around 1 km. Relative to the velocity of the meteor itself, these differences in velocity can be considered small. Therefore, we can calculate the vector velocity of the meteoroid using either group of the lateral velocity results. It is necessary to note that in the practical use of this method, the selection of antenna pairs can be flexible, and researchers should aim to choose as many channel pairs as possible while ensuring that the channel pair signals are enough strong. This ensures the accuracy and stability of the calculations.

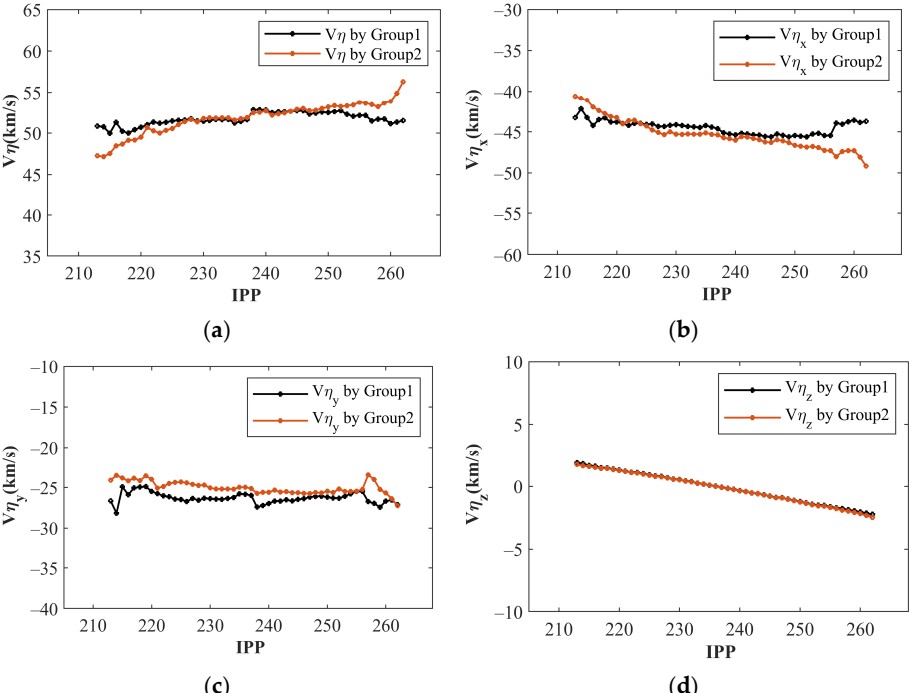

**Figure 13.** A comparison of the estimated lateral velocities between the two groups of antenna pairs. (**a**) The total lateral velocity and (**b**–**d**) its individual components in the x, y, and z directions, with the red line representing the results from the first group and the blue line representing the second group.

**Table 2.** Comparison of the total lateral velocity and three individual components in the x, y, and z directions between Group1 and Group2.

| Group1's Lateral Velocity vs. Group2's Lateral Velocity | RMSE (km/s) | MAE (km/s) |
|:---:|:---:|:---:|
| $v_\eta$ | 2.283 | 1.0361 |
| $v_{\eta_x}$ | 3.066 | 1.302 |
| $v_{\eta_y}$ | 2.355 | 1.279 |
| $v_{\eta_z}$ | 0.005 | 0.049 |

### 4.4. Result of Trajectory and Vector Velocity

Figure 14 shows the plotted trajectory and height variations of the meteoroid, respectively, while Figure 15 shows the final vector velocity measurement results. For the head echo event shown in Figure 14, our method estimated the instantaneous velocity, which was found to be consistent with the average velocity computed from the position measurements. Table 3 displays the range of calculated trajectory azimuth, elevation, height, and velocity variations. It can be observed that the meteoroid traversed the zenith and moved in the southwest direction at an altitude between 90 km and 105 km, which is consistent with the primary distribution range of meteors. Moreover, the velocity was negative in both the x and y directions, with a magnitude nearly twice the relationship, which corresponds to the motion of the meteoroid trajectory inferred from Figure 14a. Figure 15a indicates that the velocity of the meteoroid is so great that there is almost no decrease in velocity within the range of IPPs, which is noteworthy. Assuming that the meteoroid did not originate from outside the solar system or encounter third-body perturbation, the total velocity cannot exceed 72.8 km/s [25]. As this event is already very close to the upper limit, we expect further analysis of more meteor samples. If the velocity of the meteoroid is concentrated at a very high level, the radial component of the velocity vector may not be apparent, and important velocity information may be neglected in conventional radar detection. These are often crucial to research in meteor dynamics. In this article, the method proposed by us enables us to calculate the deceleration in all directions, not just in the radial direction. This provides a more detailed observing capability than previously available, and the motion characteristics of the meteoroid can be obtained using the method proposed in this study.

**Table 3.** The range of the calculated trajectory azimuth, elevation, height, and velocity variations.

| | $\theta$ (°) | $\varphi$ (°) | Height (km) | $v$ (km/s) | $v_x$ (km/s) | $v_y$ (km/s) | $v_z$ (km/s) |
|:---:|:---:|:---:|:---:|:---:|:---:|:---:|:---:|
| Begin | 46.50 | 87.80 | 102.5 | 69.25 | −39.31 | −22.72 | 51.07 |
| End | 194.5 | 87.40 | 95.30 | 71.49 | −51.13 | −27.77 | 42.12 |

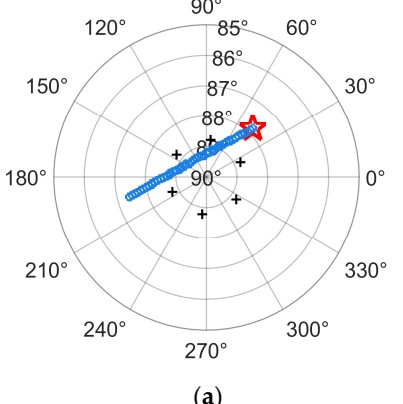

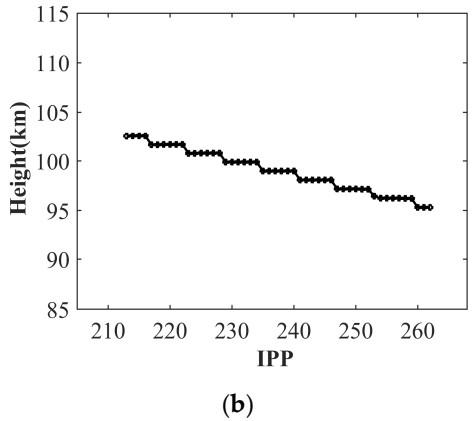

(**a**)  (**b**)

**Figure 14.** Example of a long-lasting and intense meteor head echo observed with MU on 18 June 2018 at 08:19:00 (UTC + 8/JST). (**a**) The top view of the meteor trajectory. The beginning of the event (red star) and the trajectory (blue circle). (**b**) The height variation of the meteor.

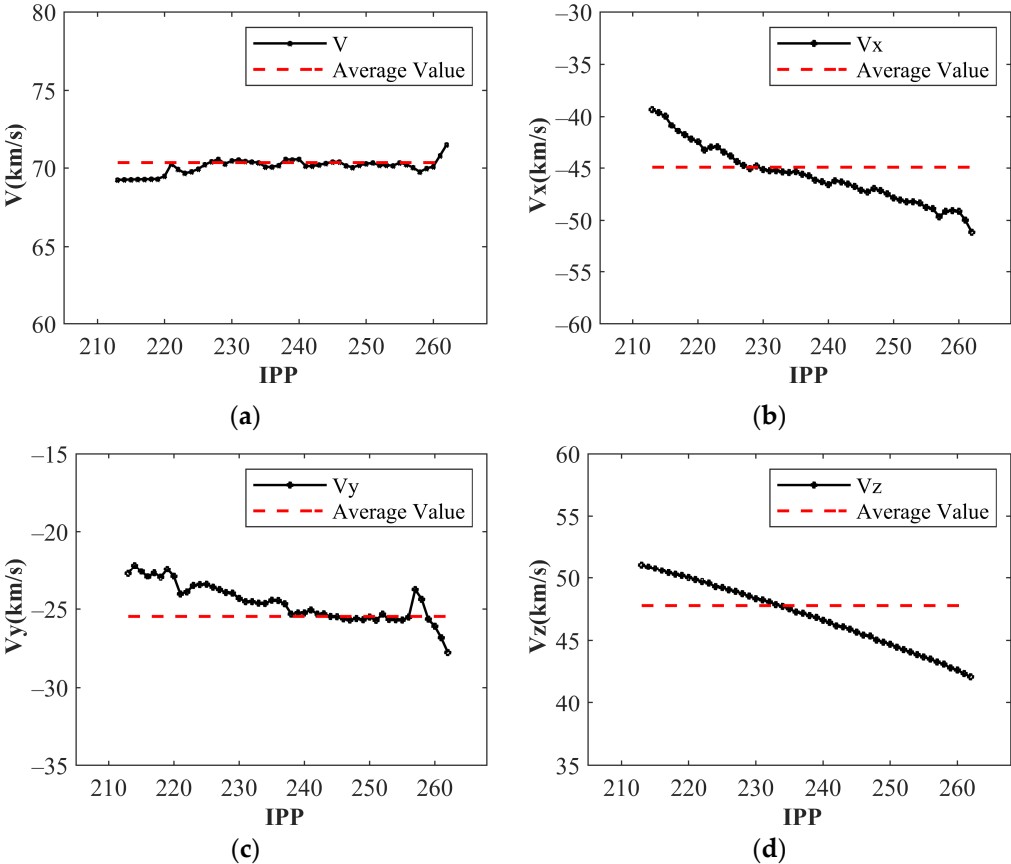

**Figure 15.** The velocity results calculated for the example shown in Figure 14. (**a–d**) The vector velocity of the meteor and its three velocity components in Cartesian coordinates. The black circle represents the instantaneous velocity of the meteoroid, while the red dashed line represents the average velocity obtained from fitting the trajectory.

## 5. Discussion and Conclusions

The proposed method in this article provides a straightforward and efficient approach to measure the vector velocity of meteoroids by utilizing meteor head echo data. By selecting multiple channel pairs and utilizing the rate of change of the phase difference, the lateral velocity and radial velocity of the meteor can be obtained, enabling the calculation of its vector velocity. The validation using MU radar head echo data showed that the calculated vector velocity was consistent with the meteor movement trajectory under the condition of self-consistency. To clarify, this method is dependent on high-precision phase estimation and therefore necessitates an adequate SNR for optimal performance. Thanks to the high power and large aperture features of the MU radar, the transmitted signal power is sufficiently strong and can form a strong transmitting beam gain, ensuring that the received echo intensity can still provide enough accuracy in extracting the phase difference. Therefore, the method can be applied to the majority of meteor head echo data. However, as observed from Figure 15, although the velocity appears relatively continuous, there are still velocity fluctuations observed on partial IPP. This is mainly attributed to the influence of SNR during the phase difference extraction process at those IPPs. Due to the dependence of phase difference on SNR, further exploration of phase difference extraction techniques is warranted in the future. Certainly, for more extreme cases of velocity estimation, such as multiple targets or prolonged interference, further research is also needed. As direct acquisition of the true parameters of meteors is unrealistic, we plan to conduct further validation of our method by comparing it with other observation techniques, such as optical cameras. We still believe that this method may have potential applicability in other detection fields. It is easy to implement, requiring only the deployment of one or multiple

pairs of interference channels. Additionally, this method does not necessitate multi-station observations, which can help save antenna channels and spatial resources. It may perhaps be applied to vector velocity observations arranged in small-area array configurations.

The method of acquiring vector velocity has advantages over conventional meteor radars, as it does not require multiple station cooperation, which can save costs and simplify the system. In addition, it determines the instantaneous velocity of each IPP, rather than the average meteor velocity obtained from the variation of travel distance over time. This is important because it implies a more detailed observation capacity, which is relevant for studying meteoroid dynamic characteristics and atmospheric properties, such as meteoroid deceleration, acceleration, explosion, or abrupt change. Furthermore, the accurate determination of meteoroid velocity is essential for determining its radiation and orbit around the sun [28]. Utilizing this method, a statistical analysis of the velocity distribution of a large number of meteors can be conducted, which can provide references for hazard estimation of spacecraft and manned space missions. With the help of this method, the past and future positions of objects can be estimated more accurately, which is crucial for tracking and determining the origin and impact points of meteoroids. We also hope that the research in this article can provide assistance for further studies in this field.

**Author Contributions:** Conceptualization, X.X.; methodology, Z.C. and X.X.; software, X.X.; validation, X.X., L.W. and Z.C.; formal analysis, X.X. and H.Z.; investigation, X.X.; resources, Z.C.; data curation, Z.C.; writing—original draft preparation, X.X.; writing—review and editing, X.X. and L.W.; visualization, X.X.; supervision, Z.C. and X.W.; project administration, Z.C. and X.W.; funding acquisition, Z.C. and X.W. All authors have read and agreed to the published version of the manuscript.

**Funding:** This research was supported in part by the National Key Research and Development Program of China, grant number 2016YFC1401100.

**Data Availability Statement:** The data presented in this study are available from the corresponding author on reasonable request.

**Acknowledgments:** The authors thank H. Hashiguchis, who is currently with the Institute of Radio Science Center for Space and Atmosphere, Kyoto University, for his help in data analysis and providing the data used in this study. The MU radar belongs to and is operated by the Radio Science Center for Space and Atmosphere, Kyoto University.

**Conflicts of Interest:** The authors declare no conflict of interest.

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
