# Peer review of "Determination of Meteor Vector Velocity Using MU Interferometry Measurements of Head Echoes"

_remotesensing, doi:10.3390/rs15153784_

Round 1

Reviewer 1 Report

Xie et al have demonstrated a method to measure the vector velocity of meteors using a single station MU radar. The proposed method is based on detecting vector velocity of head echoes. By using a single interferometric MU radar station, this method consumes less antenna and computational resources compared to multi-station methods. The data were collected from a MU radar from RASC during a 48 hours period in 2018. The results of vector velocity and trajectory from signal analysis are demonstrated. Overall, this paper is well organized, with claims well supported. I would thus recommend the acceptance of this paper with the following modification suggestions.

In the experimental setup part, the authors mentioned the power of the MU radar and the information of the antenna array. Can the author discuss how do MU radar power and antenna array number affect the accuracy of the extracted vector velocities? For example, will the result accuracy be the same if the power is reduced? And since the results from group1 and group2 are very close in Figure 13, can the author discuss if it means that the 25 sub arrays is redundant? How does the antenna number relate to the result? What is the minimum number of antenna to get a reasonable vector velocity?

In Part 4, the authors mentioned “a statistical method was chosen for phase calibration”. Can the authors elaborate on how does this method work? Also in this part, the authors should give more details on how data is selected. When selecting the data, is the strongest 86 head echoes selected or is it a random selection?

Reviewer 2 Report

I think that this manuscript is publishable but the authors need to revise the manuscript. Please see the report.

English needs slight edition.

Reviewer 3 Report

Comments: In this paper,  an approach is proposed for measuring the vector velocity of meteoroids using meteor head echoes.   1. Contributions and novelty of this paper are generic. The novelty of this work should be emphasized.   2. It is recommended to add a table that describes the quantitative results for Figures 9, 12, 13, 14 and 15 for better understanding.   3. I have one curiosity, will this study be applicable to the applications of synthetic aperture radar data?   4. Discuss in more detail with respect to the outcomes of this study. There is a lack of proper discussion in this paper.

Minor editing is required

Round 2

Reviewer 3 Report

This manuscript can be accepted for publication.

Proper minor rechecking of each word should be checked before publication.